# Systemic and Mucosal Immune Responses Induced by Adenoviral-Vectored Consensus H5 Influenza A Vaccines in Mice and Swine

**DOI:** 10.3390/vaccines13090928

**Published:** 2025-08-30

**Authors:** Adthakorn Madapong, Joshua Wiggins, Jennifer DeBeauchamp, Richard J. Webby, Eric A. Weaver

**Affiliations:** 1Nebraska Center for Virology, University of Nebraska-Lincoln, Lincoln, NE 68583, USA; amadapong2@nebraska.edu (A.M.); jwiggins9@nebraska.edu (J.W.); 2St. Jude Children’s Research Hospital, Memphis, TN 38105, USA; jennifer.debeauchamp@stjude.org (J.D.); richard.webby@stjude.org (R.J.W.); 3School of Biological Sciences, University of Nebraska-Lincoln, Lincoln, NE 68583, USA

**Keywords:** adenovirus, centralized consensus, H5Nx, HPAI, influenza A virus, vaccine

## Abstract

Background/Objectives: The continued evolution and cross-species transmission of clade 2.3.4.4b H5Nx highly pathogenic avian influenza (HPAI) viruses underscores the need for broadly protective vaccines in swine, a key intermediary host. This study aimed to evaluate systemic and mucosal immune responses elicited by adenoviral-vectored (Ad) vaccines encoding a centralized consensus hemagglutinin antigen (H5CC) in mice and swine. Methods: We constructed H5CC-based vaccines that were delivered using replication-defective (Ad5 and Ad6) and replication-competent (Ad28 and Ad48) human adenoviral vectors. Using a serotype-switched prime-boost strategy, vaccines were delivered intramuscularly (IM) or intranasally (IN) in mice and swine. We determined humoral, mucosal, and cell-mediated immune responses by hemagglutination inhibition (HI), microneutralization assay (MNA), ELISA, and IFN-γ ELISpot. Protective efficacy was evaluated by lethal H5N1 challenge in mice. Results: All vaccine strategies and routes induced significant levels of anti-H5 immunity. Ad5/Ad6 IM immunization elicited strong systemic IgG and MNA titers and robust T cell responses. IN delivery with Ad5/Ad6 induced superior mucosal IgA levels in lungs and nasal secretion. In swine, Ad5/Ad6 IM conferred the highest MNA titer and T cell responses, while the IN route enhanced mucosal IgA. The Ad28/Ad48 vaccines induced immunity in a similar pattern as compared to the Ad5/Ad6 strategy, but to a slightly lesser degree, in general. The commercial H1/H3 swine influenza vaccine failed to elicit cross-protective immunity. All H5CC vaccinated mice survived lethal H5N1 challenge without weight loss. Conclusions: Adenoviral-vectored H5CC vaccines elicit broad, cross-clade immunity with route-dependent immune profiles. IM vaccination is optimal for systemic and cellular responses, while IN delivery enhances mucosal immunity. These findings support the advancement of adenoviral platforms for influenza control in swine and pandemic preparedness.

## 1. Introduction

Highly pathogenic avian influenza (HPAI) viruses pose an ongoing global health concern due to rapid antigenic evolution, broadening host range, and high mortality rates among poultry and wild birds [1]. Since their emergence in Guangdong, China, in 1996, H5 viruses have undergone extensive genetic reassortment and antigenic drift, leading to widespread circulation of diverse H5Nx subtypes across Asia, Europe, Africa, and more recently, the Americas [2,3,4]. Among these, the clade 2.3.4.4b lineage has shown exceptional ability to cross species barriers, causing outbreaks in birds, marine and terrestrial mammals, and humans [5,6,7,8]. The recent detection of H5N1 infection in dairy cattle in the United States in 2024 highlights the unpredictability and zoonotic potential [9,10].

Swine play a central role in influenza ecology due to their susceptibility to both avian and human influenza A viruses (reviewed in [11]). This is attributed to their co-expression of α2,3- and α2,6-linked sialic acid receptors in the respiratory tract, facilitating viral reassortment [12,13]. Swine were involved in the emergence of the 2009 H1N1 pandemic virus [12,14]. Experimental data have also confirmed their permissiveness to clade 2.3.4.4b H5Nx viruses carrying mammalian adaptation markers [15,16]. Effective influenza vaccination in swine is thus vital for animal health, economic stability, and pandemic preparedness under a One Health approach.

Although whole inactivated virus vaccines (WIVs) are widely used in swine, they offer limited protection against antigenically drifted H5Nx strains and fail to elicit mucosal or T cell immunity [17]. WIVs also carry the risk of vaccine-associated enhanced respiratory disease (VAERD) upon heterologous challenge [18,19,20]. These limitations call for innovative vaccines that induce cross-protective humoral, mucosal, and cellular immunity.

Adenoviral vectors have shown promise as influenza vaccine platforms due to their potent immunogenicity and suitability for mucosal delivery [21,22,23,24]. Replication-defective adenovirus serotypes 5 (Ad5) and 6 (Ad6) vectors elicit robust systemic and cellular immune responses [25,26], while less common, replication-competent serotypes 28 (Ad28) and 48 (Ad48) may enhance mucosal targeting and antigen persistence [27]. However, their use in swine against H5Nx viruses remains unknown.

In this study, we developed a centralized consensus H5 hemagglutinin (H5CC) immuogen that localizes to the clade 1 branch of H5 influenza virus and is expressed in replication-defective (Ad5 and Ad6) and replication-competent (Ad28 and Ad48) human adenoviral vectors. Using serotype-switched prime-boost regimens administered intramuscular (IM) or intranasal (IN), we evaluated their immunogenicity in mice and swine. A 21-day interval between prime and boost was selected to allow maturation of adaptive immune responses, consistent with standard swine vaccination schedules [28,29]. We determined humoral, mucosal, and cellular immune responses to identify promising vaccine strategies for cross-clade protection against H5Nx influenza A viruses.

## 2. Materials and Methods

### 2.1. Ethics Statement

During the duration of this study, all biological procedures were reviewed and approved by the Institutional Biosafety Committee (IBC) at the University of Nebraska-Lincoln (protocol number: 619). In all experiments performed on mice, protocols approved by the Institutional Animal Care and Use Committee of the University of Nebraska-Lincoln (IACUC), protocol number: 2662, 2674 and conducted under biosafety level 2+ conditions were followed, as indicated by the Animal Welfare Act, the PHS Animal Welfare Policy, as well as the principles of the National Institutes of Health’s Guide for the Care and Use of Laboratory Animals. The female BALB/c mice were obtained from Jackson Laboratories, and all procedures were performed under isoflurane or ketamine/xylazine-induced anesthesia. Prior to use in the study, all mice were allowed to acclimate for at least one week. A variety of enrichment materials were provided, including Kim wipes, Nylabones, and a plastic hut, with temperatures ranging from 68–72 °F and a humidity level of 30 to 70%. Our animals were maintained on a light/dark cycle of 14 h and 10 h, housed in TECNIplast IVC cages with recycled paper bedding, and fed standard rodent chow. Under biosafety level 2+ conditions, all experiments on swine were approved by the University of Nebraska-Lincoln Institutional Animal Care and Use Committee, protocol number: 2167. Three to five-week-old outbred American Yorkshire pigs were purchased from Midwest Research Swine and randomly assigned to five immunization groups and housed in separate rooms in the University of Nebraska-Lincoln animal biosafety level 2+ research facility. Prior to immunization, pigs were acclimated for one week and provided unlimited access to food and water.

### 2.2. Viruses and Vaccines

Human embryonic kidney 239 (HEK293) and Madin-Darby canine kidney-London line (MDCK-Ln) cells were maintained in Dulbecco’s minimum essential medium (DMEM, HyClone™, Cytiva, Wilmington, DE, USA) supplemented with 5% fetal bovine serum (FBS, Gibco, Thermo-Fisher Scientific, Waltham, MA, USA), 1% penicillin/streptomycin (%v/v, Gibco), and cultured at 37 °C, 5% CO_2_. The following H5N1 influenza A viruses were obtained from the WHO GISRS network. These viruses included A/Vietnam/1204/2004 (Vietnam/2004), A/bar-headed goose/Qinghai/A/2005 (Goose/2005), and A/Japanese white-eye/Hong Kong/1038/2006 (White-eye/2006). The H5N1 A/bald eagle/Florida/W22-134-OP/2022 (Bald eagle/2022) and A/bovine/Ohio/B24OSU-439/2024 (Bovine/2024) were derived from St. Jude Children’s Research Hospital. The H5 virus genomes originated from reverse genetic systems with the A/Puerto Rico/8/1934 backbone, without the polybasic connecting peptide regions. All viruses were grown in 10-day-old specific pathogen-free embryonated chicken eggs and quantified by hemagglutination assay and tissue culture infective dose 50 (TCID_50_), and then stored at −80 °C for further use. For the challenge study, three H5 viruses (Vietnam/2004, Goose/2005, and Bovine/2024) were mouse-adapted through serial lung passages as previously described [30]. Briefly, BALB/c mice were intranasally inoculated with 50 μL of allantoic fluid containing H5 virus. After three days, lungs were harvested, homogenized, and centrifuged homogenate was used as the virus inoculum for the next passage. After a total of nine passages, mouse-adapted H5 viruses were grown and quantified as mentioned above.

### 2.3. Centralized Gene Construction and Adenovirus Vaccines

A consensus hemagglutinin gene for H5 influenza A viruses was previously constructed as described [31,32]. Briefly, 21 H5-HA sequences were obtained from GenBank, aligned using ClustalX v2.1, and used to generate a centralized consensus sequence (H5CC). The accession numbers for these sequences are as follows: ISDN38262, AY575869, AY555153, ISDN40341, ISDN119678, AB239125, AY555150, ISDN110940, AY651335, AY651334, ISDN118371, ISDN121986, ISDN117777, ISDN117778, ISDN49460, AJ867074, AY679514, AF084532, AF084279, AF084280, and AF046097. The resulting 568-amino-acid consensus sequence was codon-optimized for mammalian expression and synthesized by GenScript, Inc. Species C and D adenoviruses used in this study were modified, amplified, and purified as previously described [33]. Replication-defective (RD) viral vectors were generated by deleting the *E1* genes and replacing them with a CMV expression cassette containing the H5CC gene (Ad5- and Ad6-H5CC) [32]. Replication-competent (RC) viral vectors were produced by deleting the *E3* genes and inserting an expression cassette expressing the H5CC gene (Ad28- and Ad48-H5CC) [27].

### 2.4. Immune Correlates Study in Mice

Twenty-five female BALB/c mice were divided into five groups (*n* = 5/group) as following: Ad5/Ad6 IM, Ad5/Ad6 IN, Ad28/Ad48 IM, Ad28/Ad48 IN, and DPBS. Prime and boost immunizations were administered on day 0 (D0) and day 21 (D21). The Ad5/Ad6 IM group received 10^10^ viral particles (vp) of Ad5- and Ad6-H5CC intramuscularly (IM), while the Ad5/Ad6 IN groups were vaccinated intranasally (IN) with 10^10^ vp of Ad5- and Ad6-H5CC vaccines. Similarly, the Ad28/Ad48 IM and Ad28/Ad48 IN groups were vaccinated with 10^10^ vp of Ad28-H5CC and Ad48-H5CC vaccines, either IM or IN. DPBS-vaccinated controls were included for comparison (Figure 1). IM immunizations consisted of two 25 μL injections (total 50 μL) into both quadriceps using a 27-gauge insulin syringe. IN vaccinations were performed under ketamine/xylazine-induced anesthesia, with 10 μL of vaccine administered per nare (20 μL total volume).

Mice were sacrificed for blood, lung, and spleen collection after boost immunization (D35). A BD Microtainer Blood Collection Tube (Becton Dickenson, Becton, NJ, USA) was used for blood collection and serum separation for hemagglutination inhibition (HI), microneutralization assay (MNA), and ELISA testing. The lung samples were homogenized in DPBS and centrifuged at 21,000× *g* for 10 min. The lung supernatants were pretreated with 10 mM dithiothreitol followed by 10% bovine serum albumin for 1 h at 37 °C prior to use in ELISA. Splenocytes were obtained by running the spleen through a 40 mm nylon cell strainer (Fisher Scientific, Waltham, MA, USA) and lysing the red blood cells with ACK lysis buffer. For ELISpot assays, splenocytes were resuspended in cRPMI-1640 [RPMI-1640 media (HyCloneTM, Cytiva, Wilmington, DE, USA) supplemented with 10% FBS and 1% penicillin/streptomycin].

### 2.5. Protective Efficacy Against H5 Influenza A Viruses Challenge in Mice

To access protection against lethal challenge of H5 influenza viruses (Figure 2), mice were prime/boost immunized with vaccines at D0 and D21, as mentioned above. Two weeks after the boost immunization (D35), vaccinated mice were intranasally challenged with reverse-genetics(rg), mouse-adapted H5N1 influenza A viruses: Vietnam/2004, Goose/2005, and Bovine/2024 under ketamine/xylazine-induced anesthesia. All challenged mice were monitored for weight loss for two weeks, and mice that lost ≥ 25% of their initial weight were humanely sacrificed by CO_2_ asphyxiation followed by cervical dislocation.

### 2.6. Immune Correlates Study in Swine

Prior to conducting the study in pigs, a panel of H5 influenza was utilized to screen for preexisting maternally derived antibodies and confirm the seronegative status against influenza A virus. A commercial vaccine was employed as a standard-of-care vaccine to compare with common vaccine strategies used in swine production systems. Pigs were grouped (*n* = 5/group) and prime/boost vaccinated with the following vaccines on D0 and D21: Ad5/Ad6 IM, Ad5/Ad6 IN, Ad28/Ad48 IM, Ad28/Ad48 IN, FluSure XP, and DPBS. The Ad5/Ad6 IM group received 10^11^ vp of Ad5- and Ad6-H5CC vaccines via the IM route, respectively. The Ad5/Ad6 IN group was intranasally (IN) prime-boost immunized with 10^11^ vp of Ad5- and Ad6-H5CC vaccines, respectively. Similarly, the Ad28/Ad48 IM and Ad28/Ad48 IN groups were either IM or IN prime-boost immunized with 10^11^ vp of Ad28-H5CC and Ad48-H5CC, respectively. Pigs in the FluSure XP group were administered a 2 mL dose of FluSure XP^®^ Quadrivalent Swine Influenza A Virus Vaccine (Zoetis^®^, Parsippany, NJ, USA) via the IM route according to the manufacturer’s instructions and served as a commercial vaccine comparator. All vaccinated groups were compared to the DPBS-sham vaccinated control group (Figure 3). All IM immunizations were administered using a 21-gauge needle and syringe, with a dose of 1 mL/pig. IN immunizations were delivered using syringes equipped with the MAD Nasal™ Intranasal Mucosal Atomization Device (Teleflex, Morrisville, NC, USA), with a total dose of 1 mL/pig, 0.5 mL/nostril.

Blood samples and nasal swabs were collected on D21 and D35 following vaccinations. Sera were separated from whole blood using a BD Vacutainer Serum Separator Tube (Becton Dickenson) and subsequently used for hemagglutination inhibition (HI) and microneutralization assays (MNA), as well as ELISA. Peripheral blood mononuclear cells (PBMCs) were processed from whole blood stored in an EDTA-containing BD Vacutainer Tube (Becton Dickenson) according to a previously described method [34]. Briefly, whole blood samples were diluted 1:1 with DPBS and overlaid onto lymphocyte separation media (Cat# 25072CV, Corning^®^, New York, NY, USA) before centrifugation at 1000× *g* for 30 min. PBMCs were collected, washed with DPBS, and red blood cells were lysed with ACK lysis buffer. The PBMCs were resuspended in cRPMI-1640 and used for ELISpot assays. Nasal swabs were collected and stored in UniTranz-RT^®^ Universal Transport Medium (Puritan Medical Product, Guilford, ME, USA), then treated with DDT as previously mentioned before being used in ELISA.

### 2.7. Hemagglutination Inhibition (HI) Assay

Serum samples were obtained following both prime and boost immunizations from vaccinated animals. The sera were treated with a 1:3 ratio of receptor-destroying enzyme (Denka Seiken, Tokyo, Japan) at 37 °C for 20 h. Subsequently, the sera were heat-inactivated at 56 °C for 45 min and diluted 1:10 in DPBS. A two-fold serial dilution of the sera was performed in V-bottom 96-well plates, to which 8 hemagglutinin units (HAU) of influenza A virus were added to each well. The plates were incubated at room temperature for 1 h, after which 0.5% chicken red blood cells were added, and the plates were incubated for an additional hour at room temperature. Hemagglutination inhibition patterns were assessed by tilting the plates at a 45° angle and observing the formation of a teardrop in the wells.

### 2.8. Microneutralization Assay (MNA)

The sera were heat-inactivated at 56 °C for 30 min and subsequently subjected to a two-fold serial dilution in sterile 96-well U-bottom plates before the addition of 100 TCID_50_ of virus per well. Following a 1 h incubation at 37 °C in an incubator with 5% CO_2_, MDCK cells (2 × 10^5^ cells/mL) were added to the plates and cultured overnight at 37 °C with 5% CO_2_. After incubation, the plates were washed with DPBS, and DMEM supplemented with 0.002% TPCK-trypsin was added to each well. The plates were further incubated for 3 days at 37 °C with 5% CO_2_, after which 50 μL of 0.5% chicken red blood cells were added to each well and incubated at room temperature for 1 h. Hemagglutination patterns were assessed by tilting the plates at a 45° angle and observing the formation of a teardrop in the wells.

### 2.9. Antibody Responses As Measured by ELISA

Recombinant hemagglutinin (HA) proteins of H5 influenza A viruses were utilized for ELISA, including strains A/goose/Guangdong/1/1996 (Goose/1996, H5N1, Cat# 40024-V08H1, Sino Biological, Beijing, China), A/Vietnam/1194/2004 (Vietnam/2004, H5N1, Cat# 11062-V08H1, Sino Biological), A/northern pintail/Washington/40961/2014 (Northern pintail/2014, H5N8, NR-50174, Lot# 70006685, BEI resources), A/snow goose/Missouri/CC15-84A/2015 (Snow goose/2015, H5N2, NR-50651, Lot# 7003601, BEI resources), A/bald eagle/Florida/W22-134-OP/2022 (Bald eagle/2022, H5N1, NR-59476, Lot# 70072012, BEI resources) and A/dairy cattle/Texas/24-008749-001-original/2024 (Dairy cattle/2024, H5N1, NR-59816, Lot# 70072037, BEI resources). HRP-conjugated secondary antibodies used for ELISA included goat anti-mouse IgG H+L (Cat# 2984574, Lot# AP308P, Invitrogen, Waltham, MA, USA), goat anti-pig IgG H+L (Cat# APH865P, Lot# 156039, Bio-Rad), goat anti-mouse IgA (Cat# 1040-05, Lot# 3021-Y892D, Southern Biotech, Birmingham, AL, USA), and goat anti-pig IgA (Cat# AAI40P, Lot# 158413, Bio-Rad). Sera, diluted 1:100, were used to detect IgG and IgA, while DTT-treated nasal swabs and lung supernatants, diluted 1:4, were used for IgA detection.

Briefly, Immunolon 4 HBX microtiter 96-well plates (Thermo-Fisher Scientific) were coated with 200 ng/well of influenza virus H5-HA proteins in bicarbonate/carbonate coating buffer and incubated overnight at 4 °C. Plates were then blocked with 5% bovine serum albumin in DPBS + 0.1% Tween-20 (BSA/DPBS-T) for 2 h at room temperature. Samples (100 μL) of sera, nasal swabs, and lung supernatants were added to the plates in duplicate and incubated for 2 h at room temperature. Following incubation and washing, plates were incubated with a 1:5000 dilution of secondary HRP-conjugated antibodies in 2% BSA/DPBS-T for 1 h. After additional washing, 1-Step Ultra TMB-ELISA (Thermo-Fisher Scientific) was added to the plates. The reaction was stopped with 2 M sulfuric acid, and the optical density at 450 nm wavelength (OD_450_) was measured using a SpectraMax i3X multi-mode microplate reader (Molecular Devices, San Jose, CA, USA). Background absorbance from wells without sera was subtracted from the sample OD_450_ values.

### 2.10. Enzyme-Linked Immunospot (ELISpot) Assay

An interferon-**γ** (IFN-**γ**) ELISpot assay was used to evaluate the T cell response of mouse splenocytes and swine PBMCs following immunization. Total T cells were examined using pooled peptides from H5-HA peptide arrays including A/Thailand/4(SP-528)/2004 (Thailand/2004, H5N1, NR-2604, BEI) and A/bovine/Ohio/B24OSU-439/2024 (Bovine/2024, H5N1, synthesized by GenScript). Polyvinylidene 96-well difluoride-backed plates (MultiScreen-IP Filter plates, Sigma-Aldrich, MO, USA) were coated with 5 µg/mL of anti-mouse IFN**-**γ (AN18, Cat# 3321-3-250, Mabtech, Nacka Strand, Sweden) or anti-porcine IFN**-**γ (pIFN-γI, Cat# 3130-3-1000, Mabtech) monoclonal antibody (mAb) overnight at 4 °C. Plates were washed with DPBS and blocked with cRPMI-1640 for 1 h at 37 °C. Single-cell suspensions of 2.5 × 10^5^ splenocytes or PBMCs were added to each well and stimulated with 50 µL of peptide pools (5 µg/mL/peptide) and incubated overnight at 37 °C with 5% CO_2_ to allow for IFN-γ production. Concanavalin A (ConA, 5 µg/mL) and cRPMI-1640 were used for positive and negative control stimulations of splenocytes and PBMCs. After the incubation, plates were washed with DPBS-T and incubated with 1 µg/mL of biotinylated anti-mouse IFN-γ mAb (R4-6A2, Cat#3321-6-250, Mabtech) or anti-porcine IFN**-**γ mAb (P2C11, Cat# 3130-6-250, Mabtech) diluted in 1% FBS/DPBS for 1 h at room temperature. Plates were then washed with DPBS-T and incubated with 1:1000 dilution of streptavidin-alkaline phosphatase conjugated (Cat#3310-10-1000, Mabtech) diluted in 1% FBS/DPBS and incubated at room temperature for 1 h. After washing with DPBS-T, plates were developed by adding BCIP/NBT (Plus) alkaline phosphatase substrate (Thermo-Fisher Scientific). Development was stopped by washing several times in ddH_2_O. The plates were air-dried overnight before counting on an automated ELISpot plate reader (AID iSpot Reader Spectrum, AID GmbH, Strassberg, Germany). Results are normalized with a negative control and expressed as spot-forming units (SFU) per 10^6^ splenocytes or PBMCs. Responses were considered positive if there were greater than 50 SFU per million cells analyzed.

### 2.11. Statistical Analysis

All statistical analyses and data representations were carried out using GraphPad Prism 10 v10.4.2 (534). Data expressed as the mean with standard error (mean ± SEM). HI and MNA titers were log-transformed for statistical analysis. HI and MNA titers, ELISpot, and ELISA data were analyzed by one-way analysis of variance (ANOVA) with Tukey’s multiple comparisons. Survival outcomes were analyzed using the Kaplan–Meier log-rank test. A *p*-value < 0.05 was considered statistically significant.

## 3. Results

### 3.1. Construction and Quality Control for Ad-H5CC Vaccines

To develop a broadly protective vaccine targeting emerging H5Nx influenza A viruses, we constructed a synthetic consensus HA gene, designated H5CC, representative of clade 1.0 (Figure 4). The H5CC sequence was generated through a comprehensive phylogenetic analysis of HA genes from avian, swine, and mammalian H5 isolates collected between 2005 and 2010. This analysis incorporated global sequencing data spanning recent zoonotic and epizootic outbreaks. Phylogenetic analyses revealed that the H5CC sequence localized to a central position within clade 1 H5 viruses and is closely aligned with the ancestral A/goose/Guangdong (GsGd)/1996 lineage, from which contemporary GsGd-like H5Nx viruses are derived (Figure 4A). The percent identity between the viruses, vaccines, and proteins used in the study was determined (Appendix A).

The codon-optimized H5CC gene was cloned onto four modified-human adenovirus (Ad) vector platforms derived from serotypes 5 (Ad5), 6 (Ad6), 28 (Ad28), and 48 (Ad48). In all constructs, H5CC expression under a cytomegalovirus (CMV) immediate-early promoter with a polyadenylation (pA) signal was substituted in either *E1* or *E3* genes of the modified Ad vector genome. The resulting constructs were designated Ad5-H5CC, Ad6-H5CC, Ad28-H5CC, and Ad48-H5CC, respectively. As part of replication-defective engineering, Ad5-H5CC and Ad6-H5CC vectors were deleted from E1/E3 (Ad5) and E1 (Ad6) regions and replaced with H5CC expression cassettes, whereas Ad28-H5CC and Ad48-H5CC vectors retained E1 sequences while remaining replication-competent by substituting H5CC expression cassettes for E3 regions. The remaining Ad vector genomic structures, including transcriptional units (L1–L5, E2A/B) and noncoding regions, were preserved according to serotype-specific architectures (Figure 4B). Each vector was successfully rescued in HEK293 cells and verified by restriction enzyme digestion and full-length sequencing to confirm correct insertion and genome integrity.

### 3.2. Systemic and Mucosal Immune Responses in Vaccinated Mice

Functional antibody responses were assessed using HI and MNA against a diverse panel of H5 influenza A virus strains (Figure 5). HI titers consistently remained below the protective threshold of 1:40 across all vaccine groups and H5 strains. This was regardless of vaccination strategies or delivery routes. No statistically significant differences were observed in HI titers post-booster immunization (Figure 5A). In contrast, our results demonstrated substantial enhancement in the MNA titers of vaccinated mice in the Ad5/Ad6 IM group. Notably, this group exhibited significantly (*p* < 0.0001) elevated MNA titers against the Vietnam/2004 strain, with peak titers reaching a log_2_ value of 8 (1:256), surpassing the protective threshold (Figure 5B). Conversely, MNA titers in other groups against an H5 virus panel remained at baseline or lower levels.

To evaluate the systemic and mucosal immune responses in immunized mice, serum IgG, IgA, and lung IgA levels were analyzed using ELISA against HA antigens from various H5 subtype influenza A strains (Figure 6, Figure 7 and Figure 8). IgG antibody levels against H5-HA protein panels demonstrated a significant (*p* < 0.05) increase in all vaccinated groups compared to the DPBS group (Figure 6). However, no significant differences were observed in serum IgG levels across all vaccinated groups, irrespective of vaccination route or specific H5-HA protein, except for IgG antibodies against Goose/1996 and Bald eagle/2022 HA proteins (Figure 6A,E). In these cases, the Ad5/Ad6 IM group exhibited significantly (*p* < 0.05) elevated IgG levels compared to the Ad28/48 IM and Ad28/48 IN groups.

In serum-IgA ELISA against various H5-HA proteins (Figure 7), IN-vaccinated mice in the Ad5/Ad6 consistently showed the highest IgA levels across all H5-HA proteins. For the Goose/1996 HA protein (Figure 7A), the Ad5/Ad6 IN group exhibited significantly (*p* < 0.05) elevated IgA levels compared to all other groups, followed by the Ad28/Ad48 IN group. Both IM-vaccinated groups showed moderate responses, while the DPBS group had negligible levels. In Vietnam/2004 HA protein (Figure 7B), a similar pattern was observed. The Ad5/Ad6 IN group elicited the strongest IgA response, much higher than both the IM-vaccinated and the DPBS groups. The Ad28/Ad48 IN group induced a measurably but significantly lower response. For Northern pintail/2014 HA protein (Figure 7C), the Ad5/Ad6 IN group again generated robust IgA responses, significantly higher than all other groups. All remaining groups, including the Ad28/Ad48 IN, showed low IgA levels, with the DPBS group at background levels. In the case of Snow goose/2015 HA protein (Figure 7D), the Ad5/Ad6 IN group had the highest serum IgA levels, followed by moderate responses in the Ad28/Ad48 IM group. The Ad28/Ad48 IN and DPBS groups showed minimal responses without statistical differences. For Bald eagle/2022 HA protein (Figure 7E), the Ad5/Ad6 IN group once again elicited the strongest IgA response, significantly (*p* < 0.05) superior to all other vaccination groups. The Ad28/Ad48 IM group induced intermediate IgA levels, while the Ad28/Ad48 IN and DPBS groups showed significantly lower responses. Lastly, in response to the Dairy cattle/2024 HA protein (Figure 7F), only the Ad5/Ad6 IN group induced a strong IgA response. All the other groups, including both the Ad28/Ad48 regimens, exhibited significantly (*p* < 0.05) lower IgA levels than the Ad5/Ad6 IN group.

Lung IgA levels were significantly (*p* < 0.05) elevated in the Ad5/Ad6 IN group compared to all the other groups across five of the six HA antigens tested (Figure 8A–F). This group consistently exhibited the highest IgA values. The Ad28/Ad48 group also produced high lung IgA titers, although slightly lower than those observed in the Ad5/Ad6 IN group. In contrast, the Ad5/Ad6 IM and Ad28/Ad48 IM groups generated minimal IgA responses, with no significant difference compared to the DPBS control group.

### 3.3. Cross-Reactive T Cell Responses of Immunized Mice Against H5-HA Peptide Libraries

To evaluate systemic cellular immune responses elicited by adenoviral-vectored H5CC vaccines, splenocytes were isolated from immunized mice and stimulated with H5-HA-specific peptide libraries: Thailand/2004 and Bovine/2024. Interferon-gamma (IFN-γ) production was quantified via ELISpot assay to assess antigen-specific T cell responses (Figure 9). Stimulation with the Thailand/2004 H5-HA peptide library revealed that IM administration of both Ad5/Ad6 and Ad28/Ad48 vectors induced significantly (*p* < 0.05) higher frequencies of IFN-γ spot-forming units (SFU) compared to the IN-delivered counterparts and the DPBS control group (Figure 9A). Among vaccinated groups, the Ad5/Ad6 IM group elicited the most robust response, with mean SFU levels exceeding 2500 SFU/10^6^ splenocytes. The Ad28/Ad48 IM group also generated substantial T cell responses (~2400 SFU/10^6^ splenocytes). In contrast, IN immunization with either vector platform elicited significantly lower responses (~800–1000 SFU/10^6^ splenocytes) without statistical differences compared to the DPBS control group. A comparable trend was observed following stimulation with the Bovine/2024 HA peptide library (Figure 9B). The Ad5/Ad6 IM group demonstrated the highest (*p* < 0.05) T cell reactivity, with mean SFU levels surpassing 3000 SFU/10^6^ splenocytes. Increased T cell responses were observed in the Ad28/Ad48 IM group (~1600 SFU/10^6^ splenocytes) but showed no difference when compared with the other IN-vaccinated and DPBS control groups.

### 3.4. Protective Efficacy Against H5 Influenza A Virus Challenge in Mice

To evaluate the protective efficacy of heterologous adenoviral vector-based vaccines, mice were immunized with Ad5/Ad6 or Ad28/Ad48 vectors through IM or IN administration and subsequently challenged with one of three antigenically distinct H5N1 influenza A virus strains: Vietnam/2004, Goose/2005, or Bovine/2024 (Figure 10). Post-challenge outcomes were assessed by monitoring for daily weight loss and survival over a 14-day period. Mice in the DPBS group exhibited progressive and significant weight loss beginning 3 days post-challenge (DPC), with weight approaching 75% of baseline by 6–7 DPC (Figure 10A–C), justifying humane euthanasia in accordance with animal welfare guidelines. In contrast, all vaccinated groups, irrespective of vaccine or route of administration, were fully protected from clinical disease, maintaining over 95% of their initial body weight throughout the study. Consistent with the weight loss findings, survival analysis demonstrated 100% survival in all vaccinated groups across the three challenge strains (Figure 10D–F), while all the DPBS control mice succumbed to infection, with median survival times of 5-, 6-, and 5- DPC for Vietnam/2004, Goose/2005, and Bovine/2024, respectively.

### 3.5. Systemic and Mucosal Immune Responses in Vaccinated Swine

Functional antibody responses in swine were assessed using HI and MNA against a panel of antigenically diverse H5N1 influenza A virus strains. Following a prime immunization, HI titers remained uniformly low across all vaccinated and control groups, with no statistically significant differences observed between the vaccinated and DPBS control groups (Figure 11A). Booster immunization resulted in a modest but statistically significant (*p* < 0.0001) increase in HI titers in the Ad5/Ad6 IM group, specifically against the Vietnam/2004 strain, whereas all other groups failed to induce sufficient HI responses (Figure 11B). Despite this increase, none of the vaccine groups, including the Ad5/Ad6 IM, reached the standard protective threshold of 1:40. Notably, the commercial FluSure XP vaccine did not elicit detectable HI antibodies against any of the tested strains.

Consistent with the HI titers, MNA titers were low across all groups after prime immunization, without significant differences among groups or tested strains (Figure 11C). However, a marked increase in MNA titers was observed after boost immunization in the Ad5/Ad6 IM group. This group exhibited significantly (*p* < 0.0001) elevated MNA titers against the Vietnam/2004, Goose/2005, and White-eye/2006 H5N1 strains relative to the other groups (Figure 11D). Notably, only the Ad5/Ad6 IM group achieved a MNA titer exceeding the protective threshold, with a peak titer of 1:160 against the Vietnam/2004 strain.

To evaluate the systemic and mucosal immune responses in vaccinated pigs, serum IgG, IgA, and nasal swab IgA levels were analyzed using ELISA against HA antigens from various H5 subtype influenza A strains (Figure 12, Figure 13 and Figure 14). Following prime immunization, moderate antigen-specific IgG responses were observed across most vaccinated groups, with IM delivery of Ad5/Ad6 and Ad28/Ad48 vectors eliciting significantly (*p* < 0.05) higher serum IgG levels than their IN routes (Figure 12A–F, left panel). In contrast, the inactivated commercial vaccine (FluSure XP) induced only limited cross-reactive IgG responses, with optical density (OD_450_) values statistically comparable to the DPBS control across H5-HA antigens.

Booster immunization markedly enhanced IgG titers in all adenoviral-vectored vaccinated groups. Among these, the Ad5/Ad6 IM group consistently induced the highest IgG responses across all six HA proteins, with OD_450_ values nearing or exceeding 3.0 for Goose/1996, Vietnam/2004, Snow goose/2015, and Dairy cattle/2024 HA proteins (Figure 12A,B,D,F, right panels). The response to Northern pintail/2014 HA was also elevated but peaked closer to an OD_450_ of 2.0 (Figure 12C). Similarly, the Ad28/Ad48 IM group generated strong and statistically comparable IgG responses following boosting. While IN delivery of both adenoviral-vector vaccines resulted in lower antibody levels overall, they remained significantly (*p* < 0.05) higher than those induced in the FluSure XP or DPBS groups. Notably, FluSure XP failed to elicit detectable IgG responses against the most recent H5N1 strains, including Bald eagle/2022 and Dairy cattle/2024 (Figure 12E,F).

For serum IgA, following prime immunization, all adenoviral-vectored vaccine groups induced detectable serum IgA responses across most HA antigens tested (Figure 13A–F, left panel). Although the Ad5/Ad6 IN group exhibited numerically higher titers against certain H5-HA proteins, no statistically significant differences were observed among the adenoviral-vectored groups. However, all adenoviral vaccines elicited significantly (*p* < 0.05) higher IgA levels compared to the FluSure XP and DPBS groups, which remained at lower levels across all antigens.

Following boost immunization, IgA responses increased across all vaccinated groups, with the greatest enhancement observed in groups receiving IN administration (Figure 13A–F, right panel). The Ad5/Ad6 IN group consistently elicited the highest (*p* < 0.05) post-boost serum IgA levels across all six H5-HA antigens, including more recent strains such as Bald eagle/2022 and Dairy cattle/2024 (Figure 13E,F). The Ad28/Ad48 IN group also demonstrated significant IgA induction post-boost, although the titers were lower compared to those in the Ad5/Ad6 IN group. In contrast, the IM-vaccinated groups exhibited only a modest increase in IgA responses following boosting, remaining significantly less effective than the IN routes. The FluSure XP vaccine failed to induce relevant IgA responses at each time point, indicating its limited capacity to stimulate antibody responses.

Following prime immunization, nasal IgA responses were low or undetectable in most groups, with the notable exception of the Ad5/Ad6 IN group, which elicited significantly elevated IgA titers against several H5-HA antigens (Figure 14A–F, left panel). The highest mucosal IgA responses were observed against Vietnam/2004, Goose/1996, and Bald eagle/2022 HA proteins, with titers significantly (*p* < 0.05) higher than those induced by the IM-vaccinated, FluSure XP, or DPBS groups. The Ad28/Ad48 IN group induced low but measurable nasal IgA responses post-prime, while all IM-vaccinated groups failed to generate detectable mucosal IgA responses.

Following boosting immunization, nasal IgA responses were significantly enhanced, particularly in the IN-vaccinated group. The Ad5/Ad6 IN group remained the most effective, inducing the strongest and broadest mucosal IgA responses across all six H5-HA antigens (Figure 14A–F, right panel). Although nasal IgA titers in the Ad28/Ad48 IN group also increased post-boost, they were generally lower in magnitude and more variable than in the Ad5/Ad6 IN group. In contrast, the IM-vaccinated groups with either the Ad5/Ad6 IM or Ad28/Ad48 IM groups, as well as the commercial FluSure XP vaccine, failed to elicit significant nasal IgA responses at each time point. Nasal IgA levels from the DPBS group remained unchanged throughout the study.

### 3.6. Cross-Reactive T Cell Responses of Immunized Swine Against H5-HA Peptide Libraries

To evaluate systemic cellular immune responses elicited by adenoviral-vectored H5CC vaccines, PBMCs were isolated from vaccinated pigs post-prime and post-boost. Cells were stimulated with H5-HA-specific peptide libraries: Thailand/2004 and Bovine/2024. Interferon-gamma (IFN-γ) production was quantified via ELISpot assay to assess antigen-specific T cell responses (Figure 15). Following prime immunization, T cell responses were minimal across all groups, with only modest IFN-γ SFU levels detected in the Ad5/Ad6 IM and Ad28/Ad48 IM groups (Figure 15A,B, left panel). Responses to both peptide libraries in these groups were slightly elevated compared to controls, but did not reach statistical significance. IN-vaccinated groups (Ad5/Ad6 IN and Ad28/Ad48 IN), the commercially inactivated vaccine FluSure XP, and the DPBS groups failed to elicit noticeable T cell responses after primary immunization.

After booster immunization, a marked enhancement in HA-specific T cell responses was observed, particularly in pigs immunized via the IM route. The Ad5/Ad6 IM group demonstrated the most robust responses, with mean SFUs exceeding 2000 and 1500 per 10^6^ PBMCs against the Thailand/2004 and Bovine/2024 peptide pools, respectively (Figure 15A,B, right panel). These responses were significantly (*p* < 0.001) higher than those induced by the other groups. The Ad28/Ad48 IM group also generated elevated T cell responses post-boost, with statistically significant (*p* < 0.05) increases compared to IN-vaccinated and DPBS groups, though those responses remained lower than those elicited by the Ad5/Ad6 IM group. In contrast, boosting after IN-immunization did not significantly enhance T cell responses. Additionally, neither the FluSure XP nor DPBS groups induced significant T cell responses.

## 4. Discussion

The continued global circulation and rapid antigenic evolution of clade 2.3.4.4b H5Nx highly pathogenic avian influenza (HPAI) viruses pose a profound and escalating threat to both animal and human health. Since the emergence of these viruses in 1996, genetic drift and reassortment events have given rise to a multitude of antigenic variants that circulate in wild birds, domestic poultry, and a growing range of mammalian hosts, including swine, feline, marine mammals, and even dairy cattle in North America [4,18,35]. The risk of zoonotic spillover is also enhanced by the expansion of hosts and host ranges of H5Nx viruses, emphasizing their exceptional plasticity.

Swine occupy a pivotal niche in the influenza environment, as their respiratory epithelium co-expresses both α2,3- and α2,6-linked sialic acid receptors. This renders pigs susceptible to infection with avian and human influenza A viruses and facilitates the generation of reassortant progenies with pandemic potential [36]. The 2009 H1N1 “swine flu” pandemic serves as a stark reminder of this risk, arising through reassortment among swine, avian, and human viruses [37]. Experimental virus challenge studies further confirmed that pigs could support replication of clade 2.3.4.4b H5Nx viruses, including variants harboring mammalian adaptation markers. Consequently, pigs serve as both amplifiers and mixing vessels for novel influenza A viruses [14,38].

Commercial whole-inactivated virus (WIV) vaccines, such as FluSure XP^®^, remain the standard of swine influenza control in many production systems. However, multiple studies have documented their strain-specificity and limited cross-protection against antigenically drifted or reassorted viruses [17,18]. In this study, Flusure XP^®^ served as a comparator and consistently elicited weak immune responses, with HI and MNA titers below protective thresholds and only limited systemic and mucosal antibody inductions. Notably, Flusure XP^®^ failed to stimulate a detectable T cell response, highlighting its narrow immunological profile. Conversely, adenoviral-vectored vaccines have induced potent antibody responses across diverse H5 antigens, as well as strong IFN-γ-secreting T cell responses. These findings reinforce the limitations of conventional WIVs and support adenoviral platforms as promising alternatives for broad and durable swine influenza protection. WIVs principally elicit systemic IgG responses but fail to induce robust mucosal IgA or cellular immunity. These immune responses are critical for preventing initial infection and limiting viral shedding in the respiratory mucosa [39]. Furthermore, the risk of vaccine-associated enhanced respiratory disease (VAERD), characterized by exacerbated lung pathology upon heterologous challenge, has been linked to non-neutralizing antibody responses induced by WIVs in pigs [18,20]. Additionally, the need for constant reformulation and re-vaccination to match circulating strains further limits the practicality and cost-effectiveness of WIV programs in dynamic production environments [40].

In contrast, adenoviral-vectored vaccines offer several unique advantages. They can deliver transgenes encoding antigens and elicit potent, balanced humoral and cellular responses without adjuvants. Replication-defective human Ad5 and Ad6 have been extensively characterized and shown to induce durable neutralizing antibody titers and strong CD4^+^/CD8^+^ T cell immunity in multiple species, including swine, nonhuman primates, and humans [41,42,43]. Moreover, rare serotypes such as Ad28 and Ad48, which retain replication competence, may afford prolonged antigen expression and enhanced mucosal tropism [27,33]. It is noteworthy that we applied a consensus centralized HA antigen (H5CC) that was developed nearly 20 years ago as an H5 influenza vaccine against intermediate and contemporary H5 strains. The H5CC antigen exhibited strong immunological cross-reactivity with contemporary field strains, including recent zoonotic H5N1 viruses such as Bald eagle/2022 and Dairy cattle/2024. As a result of this long-term durability, consensus-based antigen designs provide a robust solution for avoiding antigenic drift and maintaining long-term efficacy.

Our data demonstrate that Ad5/Ad6 vectors expressing H5CC, administered via IM prime–boost regimens, generated the highest systemic HI and MNA titer in pigs, surpassing the protective threshold of 1:40 against H5N1 strains such as Vietnam/2004 and Goose/2005. HI and MNA provide functional correlates of protection, assessing antibody-mediated inhibition of virus infection [44,45]. Despite the induction of high HA-specific immunoglobulin responses in the vaccinated groups, functional HI and MNA titers remained undetectable. This observation highlights the complexity of humoral immunity, where binding antibody responses do not directly correlate with neutralization potential; several mechanisms may account for this discrepancy. First, the induction of antibodies to non-neutralizing or subdominant epitopes close to the receptor-binding site (RBS) or fusion peptide, which lack inhibitory function [46]. Second, ELISA often uses recombinant HA in a non-native conformation, which could overestimate functionally relevant antibody titers. Third, antibodies with high binding avidity may possess insufficient affinity or fail to block receptor engagement. Previous quantitative models indicate that HI titers are determined by both antibody concentration and avidity, and that low avidity can result in low or undetectable HI despite significant IgG levels [47]. Moreover, certain influenza viruses, particularly H5 subtypes, might have poor hemagglutination properties or atypical receptor-binding characteristics, reducing assay sensitivity [48,49].

Additionally, the viruses used in our assays were propagated from embryonated eggs. It is possible that this may not fully replicate natural immune response to infection. There are limitations to these serological assays, particularly when the type of erythrocyte used affects the hemagglutination pattern. Additionally, the viral HA exhibits species-specific differences in its ability to hemagglutinate erythrocytes, which can impact hemagglutination inhibition [50,51]. For influenza protection, antibodies are not only responsible for neutralization but also play a role in antibody-dependent cellular cytotoxicity (ADCC) and antibody-dependent phagocytosis (ADP) [52,53]. These protective mechanisms are currently being investigated. Importantly, IM vaccination also elicited strong IFN-γ–producing T cells upon stimulation with H5-HA peptide pools from divergent strains (Thailand/2004 and Bovine/2024), consistent with the critical role of cellular immunity in heterosubtypic protection [54]. These results are consistent with prior studies demonstrating that centralized or computationally optimized HA antigens can elicit broad immunity across divergent influenza lineages [32,34,41]. Despite decades of viral evolution, the H5CC antigen is considered an effective universal antigen for the next generation of swine and pandemic influenza vaccines.

A major innovation of this study is the demonstration that IN administration of Ad5/Ad6 vectors markedly enhances mucosal immune responses. IN delivery induces significantly elevated IgA titers in nasal secretions, lung supernatants, and serum against a broad panel of H5Nx antigens, including recent isolates Bald eagle/2022 and Dairy cattle/2024. These data corroborate the findings in pigs and poultry, where IN vaccination with the Ad5 vector produces superior local IgA and reduces viral shedding compared with parenteral routes [55,56]. Mucosal IgA neutralizes virus at the portal of entry, limiting replication and curbing transmission, attributes essential for controlling outbreaks in high-density swine production settings. While Ad28 and Ad48 vectors induced measurable responses, their immunogenicity, particularly via IN delivery, was inferior to that of Ad5/Ad6. This may reflect differences in innate tropism, antigen expression kinetics, or vector preexisting immunity in swine populations. The exploration of alternative serotypes or capsid-chimeric constructs may circumvent anti-vector immunity and broaden its applicability in field-exposed herds [57,58].

Despite promising immunogenicity, several critical gaps remain. First, protective efficacy must be validated in live-virus challenge trials in pigs. This is to assess clinical protection, viral load reduction, and transmission dynamics under conditions that mimic commercial farming. Second, immunity durability, especially mucosal IgA and tissue-resident memory T cells, warrants long-term monitoring to define optimal booster intervals. Third, the impact of maternal antibodies on vaccine take in neonatal pigs should be evaluated, as maternally derived IgG can interfere with parenteral vaccines but may be bypassed by mucosal delivery [59]. Despite maternal antibodies, adenoviral-vectored vaccines can transduce mucosal epithelial cells and antigen-presenting cells locally, triggering a mucosal IgA response and priming tissue-resident T cells. Therefore, they are suitable candidate for neonatal swine vaccination strategies. Fourth, field trials under varied management systems are essential to understand real-world efficacy, scalability, and cost–benefit considerations, including formulation stability and delivery modes. Next, heterologous prime–boost regimens, combining IM and IN routes or different vector platforms, may synergize systemic and mucosal immunity to achieve sterilization protection (reviewed in [60]).

## 5. Conclusions

In conclusion, our findings underscore the versatility and potency of adenoviral-vectored H5CC vaccines for swine influenza control. Ad5/Ad6 vectors, delivered intramuscularly for systemic immunity or intranasally for mucosal defense, elicit comprehensive immune responses with cross-clade breadth. Addressing the identified limitations through challenge studies, field evaluations, and formulation enhancements will be vital for advancing these platforms toward practical, field-deployable vaccines that underpin One Health strategies to mitigate influenza transmission at the animal–human interface.

## Figures and Tables

**Figure 1 vaccines-13-00928-f001:**
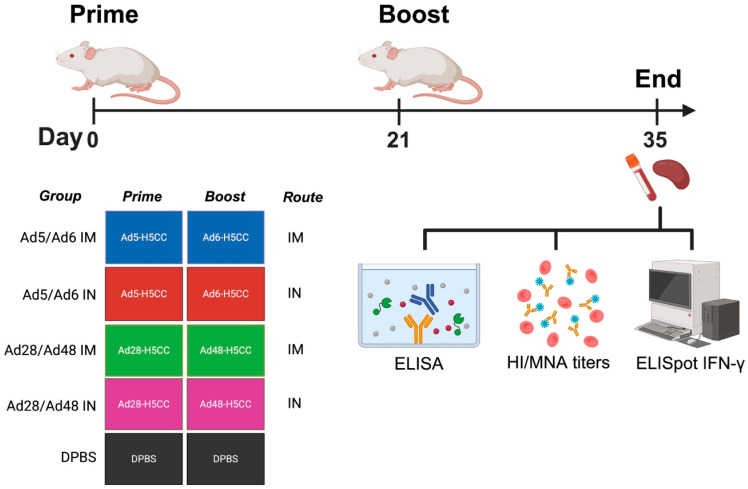
Schematic illustration of immunological correlates in mice. Groups of mice (*n* = 5/group) were immunized with different Ad-H5CC vaccines intramuscularly (IM) or intranasally (IN) on days 0 (D0) and 21 (D21). Samples were collected at D35 and used to evaluate the humoral immune response via hemagglutination inhibition (HI), microneutralization assay (MNA), and ELISA. Cell-mediated immune response was evaluated using IFN-γ ELISpot assay.

**Figure 2 vaccines-13-00928-f002:**
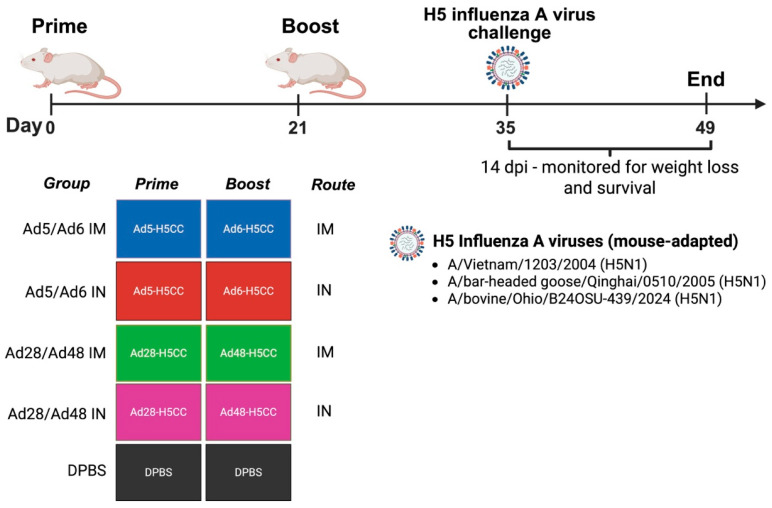
Schematic illustration of a protective efficacy study against H5 influenza A virus challenge in mice. Groups of mice (*n* = 5/group) were immunized with different Ad-H5CC vaccines by the IM or IN routes at D0 and D21. At D35, mice were challenged with rgH5 influenza A viruses. Weight loss and survival were recorded and analyzed.

**Figure 3 vaccines-13-00928-f003:**
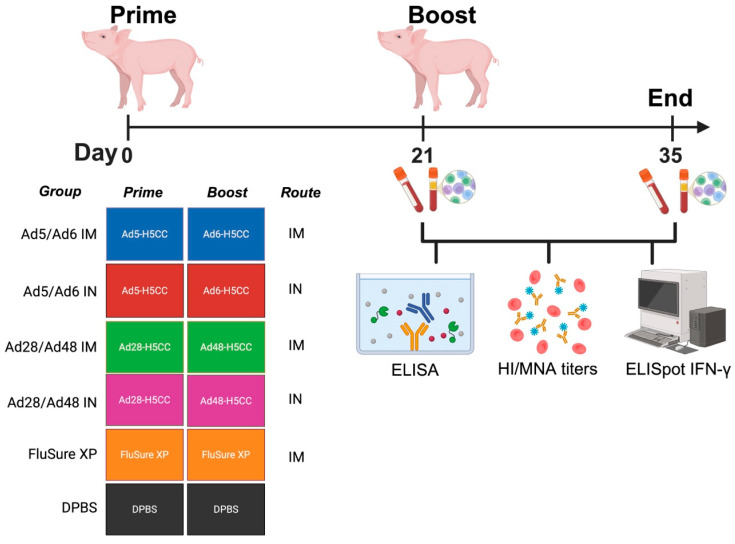
Schematic illustration of immune correlates in swine. Groups of pigs (*n* = 5/group) were prime-boost vaccinated with different Ad-H5CC vaccines either by the IM or IN routes at D0 and D21. Samples were collected at D21 and D35 and used to evaluate for humoral immune response using HI, MNA, and ELISA. Cell-mediated immune response was evaluated using IFN-γ ELISpot assay.

**Figure 4 vaccines-13-00928-f004:**
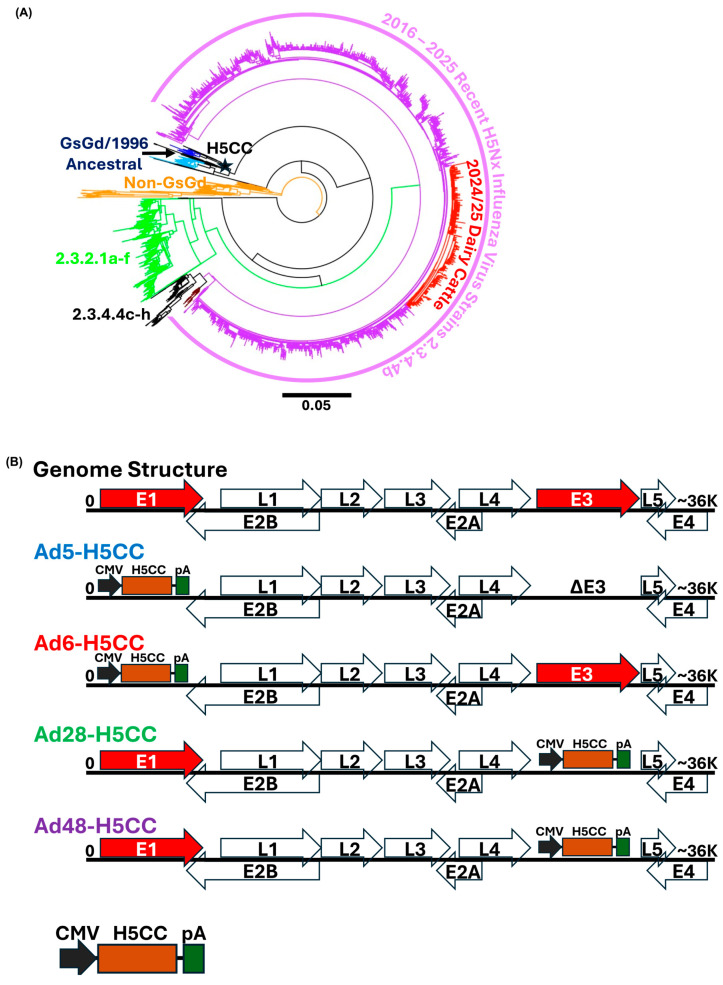
Construction of adenoviral-vectored vaccines used in this study. (**A**) Phylogenetic analysis of H5 influenza A viruses highlighting the position of H5CC gene (star), which clusters centrally within the clade 1 viruses and is closely related to the ancestral GsGd/1996 lineage. (**B**) Schematic representation of adenoviral vector backbones used for H5CC vaccine delivery. Four human adenovirus serotypes, Ad5 and Ad6 (replication-defective) and Ad28 and Ad48 (replication-competent), were engineered to express the H5CC gene under the control of a CMV promoter and a polyadenylation (pA) signal (**bottom panel**). In Ad5 and Ad6 vectors, the H5CC cassette replaces the deleted E1 region, with Ad5 also containing an E3 deletion. In Ad28 and Ad48 vectors, the H5CC cassette replaces the E3 region while retaining the E1 region.

**Figure 5 vaccines-13-00928-f005:**
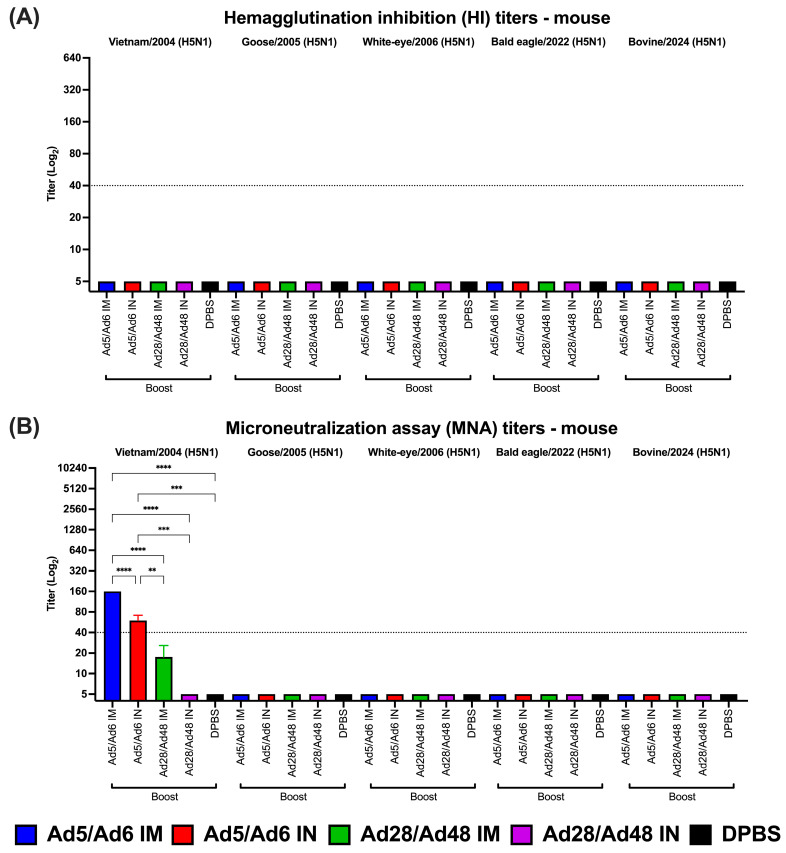
(**A**) Hemagglutination inhibition (HI) and (**B**) microneutralization assay (MNA) titers of immunized mice against H5 influenza A viruses. Values are expressed as mean ± SEM. Dashed lines indicate a 50% protective titer (≥1:40; 5.32 log_2_). Asterisks indicate statistical differences among groups (**** *p* < 0.0001; *** *p* ≤ 0.0008; ** *p* ≤ 0.002).

**Figure 6 vaccines-13-00928-f006:**
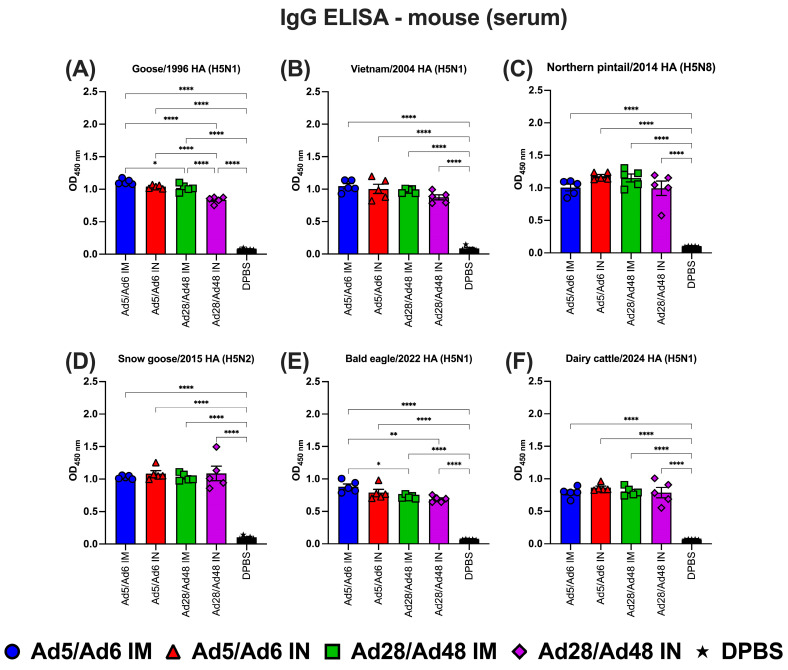
Immunoglobulin G (IgG) in mouse serum against H5-HA proteins panel: (**A**) Goose/1996 HA (H5N1), (**B**) Vietnam/2004 HA (H5N1), (**C**) Northern pintail/2014 HA (H5N8), (**D**) Snow goose/2015 HA (H5N2), (**E**) Bald eagle/2022 HA (H5N1), and (**F**) Dairy cattle/2024 HA (H5N1). Values are expressed as mean *±* SEM. Asterisks indicate statistically significant differences among groups (**** *p* < 0.0001; ** *p* ≤ 0.002; * *p* ≤ 0.05).

**Figure 7 vaccines-13-00928-f007:**
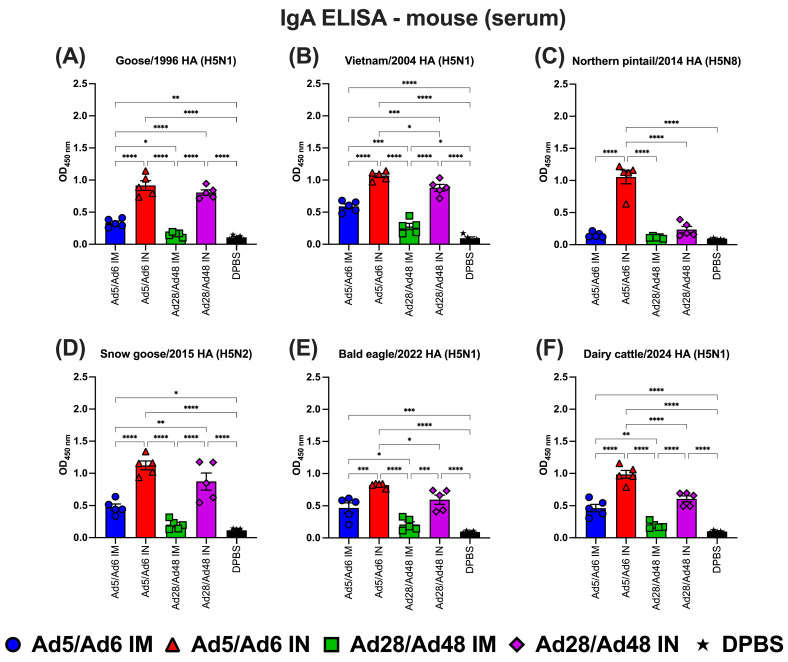
Immunoglobulin A (IgA) in mouse serum against H5-HA proteins panel: (**A**) Goose/1996 HA (H5N1), (**B**) Vietnam/2004 HA (H5N1), (**C**) Northern pintail/2014 HA (H5N8), (**D**) Snow goose/2015 HA (H5N2), (**E**) Bald eagle/2022 HA (H5N1), and (**F**) Dairy cattle/2024 HA (H5N1). Values are expressed as mean *±* SEM. Asterisks indicate statistically significant differences among groups (**** *p* < 0.0001; *** *p* = 0.0002; ** *p* = 0.007; * *p* ≤ 0.05).

**Figure 8 vaccines-13-00928-f008:**
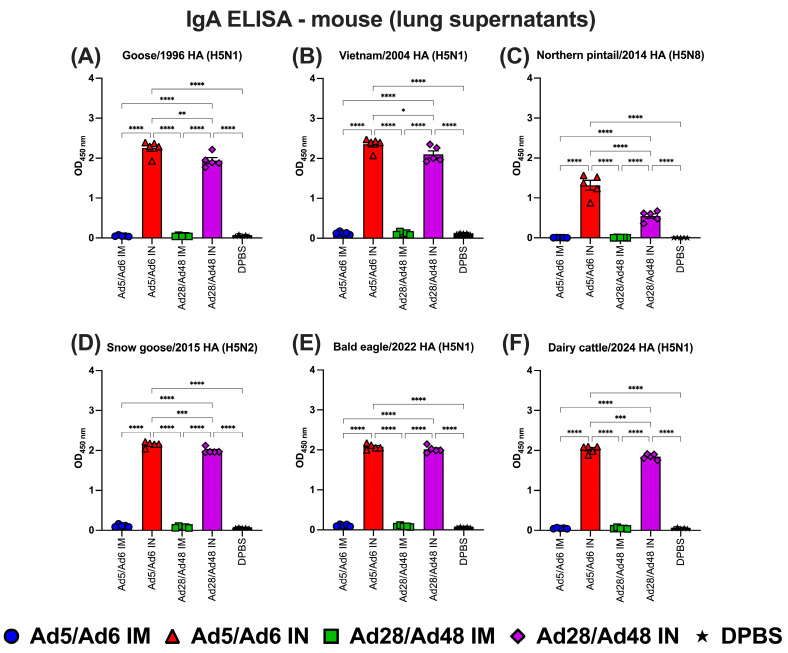
Immunoglobulin A (IgA) in lung supernatants of immunized mice against H5-HA proteins panel: (**A**) Goose/1996 HA (H5N1), (**B**) Vietnam/2004 HA (H5N1), (**C**) Northern pintail/2014 HA (H5N8), (**D**) Snow goose/2015 HA (H5N2), (**E**) Bald eagle/2022 HA (H5N1), and (**F**) Dairy cattle/2024 HA (H5N1). Values are expressed as mean *±* SEM. Asterisks indicate statistically significant differences among groups (**** *p* < 0.0001; *** *p =* 0.0002; ** *p* ≤ 0.002; * *p* ≤ 0.05).

**Figure 9 vaccines-13-00928-f009:**
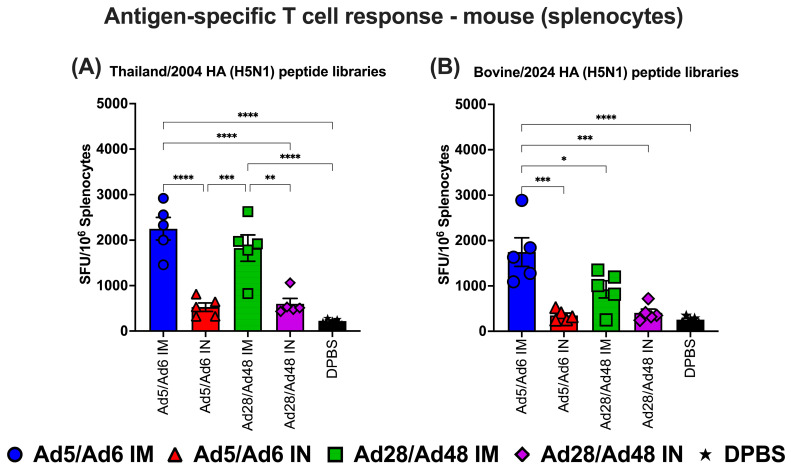
Antigen-specific T cell responses in splenocytes from immunized mice against H5-HA peptide libraries: (**A**) Thailand/2004 and (**B**) Bovine/2024. Values are expressed as mean ± SEM. Asterisks indicate statistically significant differences among groups (***** p* < 0.0001; **** p* = 0.0001; *** p* ≤ 0.002; * *p* ≤ 0.05).

**Figure 10 vaccines-13-00928-f010:**
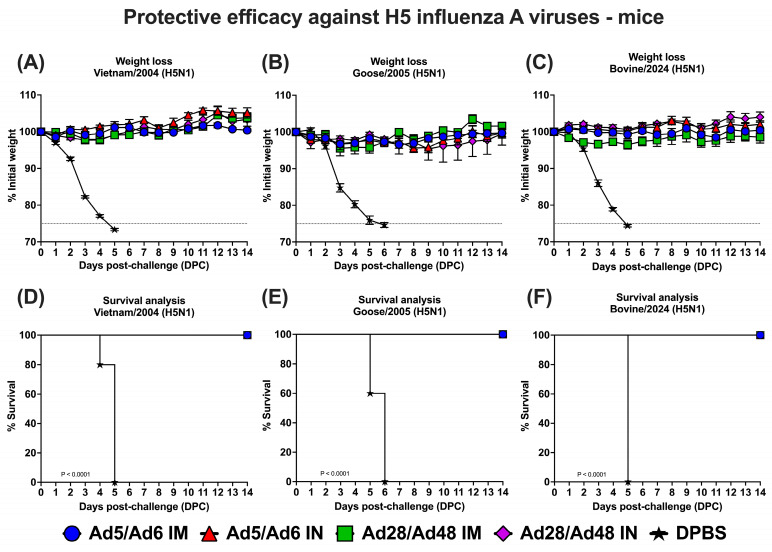
Protection against lethal H5N1 influenza A viruses. Groups of mice (n = 5/group) were immunized with different Ad-H5CC vaccines by either the IM or IN routes at D0 and D21. At D35, mice were challenged with mouse-adapted influenza viruses: (**A**,**D**) A/Vietnam/1203/2004 (Vietnam/2004 H5N1), (**B**,**E**) A/bar-headed goose/Qinghai/A/2005 (Goose/2005 H5N1), and (**C**,**F**) A/bovine/Ohio/B24OSU-439/2024 (Bovine/2024 H5N1). Mice were monitored for weight loss and survival. Dashed lines in weight loss curves indicate a threshold for ≥25% weight loss.

**Figure 11 vaccines-13-00928-f011:**
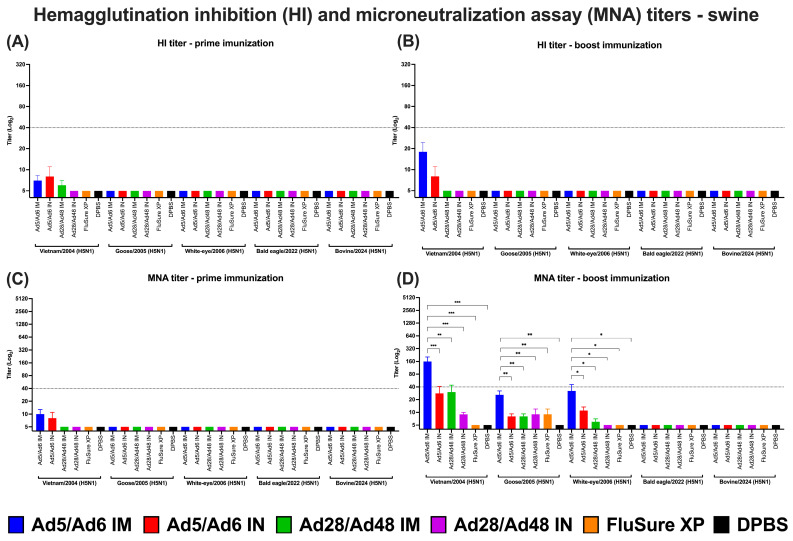
(**A**,**B**) Hemagglutination inhibition (HI) and (**C**,**D**) microneutralization assay (MNA) titers of vaccinated pigs against H5 influenza A virus panels. Values are expressed as mean ± SEM. Dashed lines indicate a 50% protective titer (≥1:40; 5.32 log_2_). Asterisks indicate statistically significant differences among groups (**** p* = 0.0001; *** p* ≤ 0.002; ** p* ≤ 0.05).

**Figure 12 vaccines-13-00928-f012:**
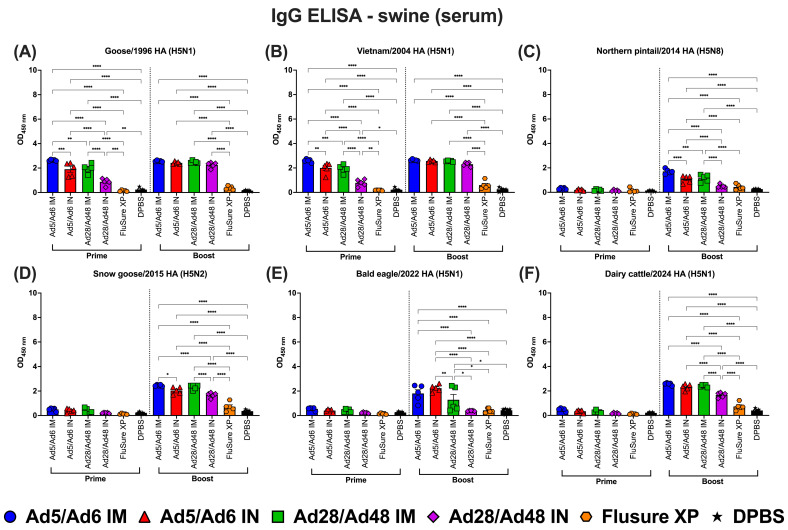
Immunoglobulin G (IgG) in swine serum against H5-HA proteins panel: (**A**) Goose/1996 HA (H5N1), (**B**) Vietnam/2004 HA (H5N1), (**C**) Northern pintail/2014 HA (H5N8), (**D**) Snow goose/2015 HA (H5N2), (**E**) Bald eagle/2022 HA (H5N1), and (**F**) Dairy cattle/2024 HA (H5N1). Values are expressed as mean *±* SEM. Asterisks indicate statistically significant differences among groups (**** *p* < 0.0001; *** *p* = 0.0001; ** *p* ≤ 0.002; * *p* ≤ 0.05).

**Figure 13 vaccines-13-00928-f013:**
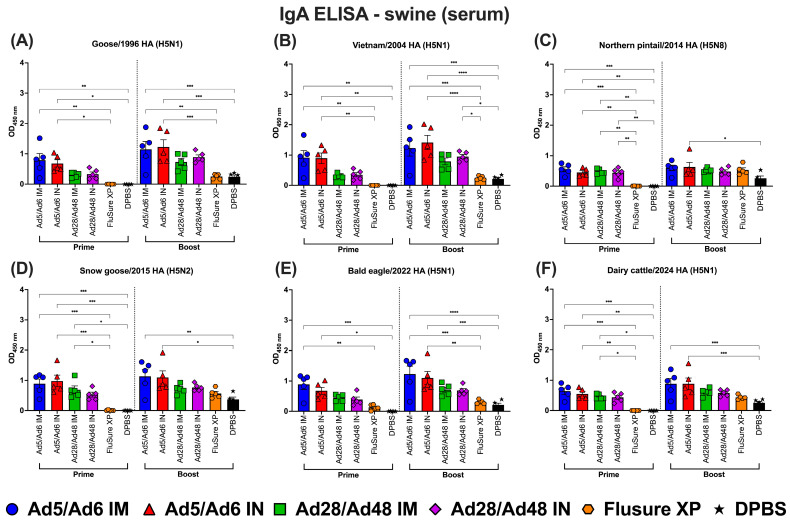
Immunoglobulin A (IgA) in swine serum against H5-HA proteins panel: (**A**) Goose/1996 HA (H5N1), (**B**) Vietnam/2004 HA (H5N1), (**C**) Northern pintail/2014 HA (H5N8), (**D**) Snow goose/2015 HA (H5N2), (**E**) Bald eagle/2022 HA (H5N1), and (**F**) Dairy cattle/2024 HA (H5N1). Values are expressed as mean ± SEM. Asterisks indicate statistically significant differences among groups (***** p* < 0.0001; **** p* = 0.0001; *** p* ≤ 0.002; ** p* ≤ 0.05).

**Figure 14 vaccines-13-00928-f014:**
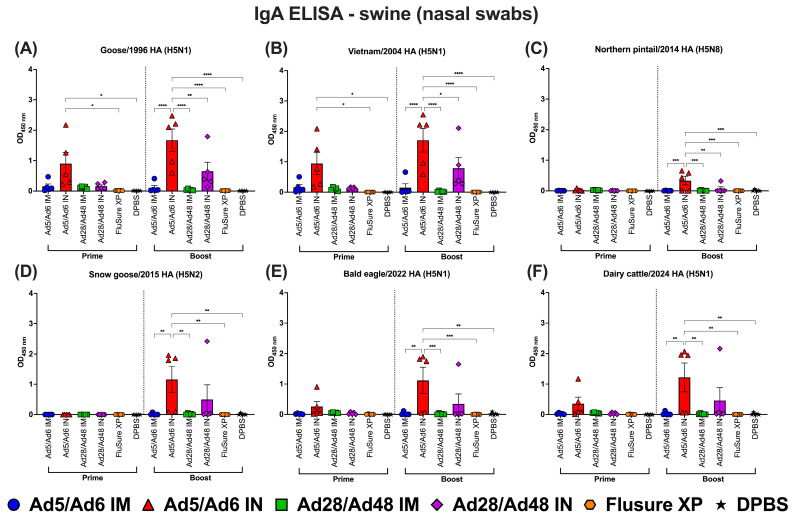
Immunoglobulin A (IgA) in nasal swabs of swine against H5-HA proteins panel: (**A**) Goose/1996 HA (H5N1), (**B**) Vietnam/2004 HA (H5N1), (**C**) Northern pintail/2014 HA (H5N8), (**D**) Snow goose/2015 HA (H5N2), (**E**) Bald eagle/2022 HA (H5N1), and (**F**) Dairy cattle/2024 HA (H5N1). Values are expressed as mean ± SEM. Asterisks indicate statistically significant differences among groups (***** p* < 0.0001; **** p* = 0.0001; *** p* ≤ 0.002; ** p* ≤ 0.05).

**Figure 15 vaccines-13-00928-f015:**
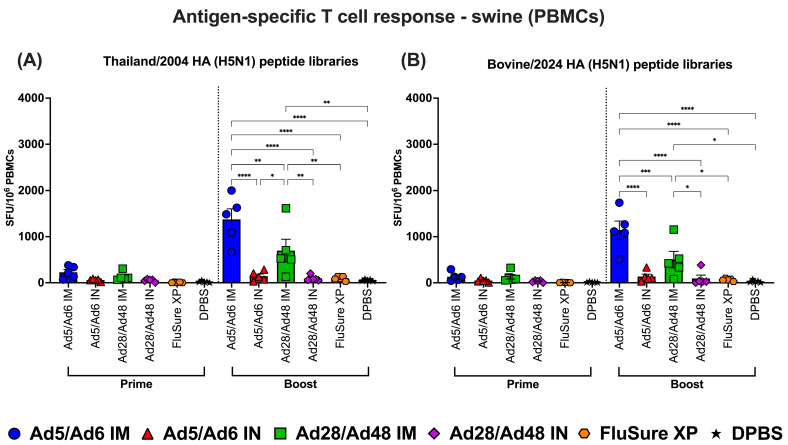
Antigen-specific T cell responses in splenocytes from vaccinated swine against H5-HA peptide libraries: (**A**) Thailand/2004 and (**B**) Bovine/2024. Values are expressed as mean ± SEM. Asterisks indicate statistically significant differences among groups (***** p* < 0.0001; **** p* = 0.0001; *** p* ≤ 0.002; ** p* ≤ 0.05).

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
