# Peer review of "Systemic and Mucosal Immune Responses Induced by Adenoviral-Vectored Consensus H5 Influenza A Vaccines in Mice and Swine"

_vaccines, 2025, doi:10.3390/vaccines13090928_

Round 1
Reviewer 1 Report
Comments and Suggestions for Authors
This manuscript presents the results of an extensive experiment to evaluate the immunogenicity and protective activity of broad-spectrum vaccines against H5Nx viruses generated in this work on the basis of adenoviral platforms, both replicatively effective and replicatively defective. The strength of the work lies in the fact that for animal immunization, along with homologous prime-boost vaccination, the heterologous prime-boost approach was used. Vaccine prototypes were studied in both mouse and pig models, and both humoral and T-cell responses to immunization were evaluated. The vaccine antigens contained the consensus protein HA, which did not lead to the formation of HAI antibodies to various H5 clades of influenza viruses. Nevertheless, during heterologous prime boost immunization, cross-reactive neutralizing antibodies were formed, as well as binding antibodies detected in ELISA.
The paper is written very well, and the authors also discuss the shortcomings of their research. In general, the research is important and it needs to be published; I have only one small remark:
Figures 6-9 and 11-15. Please change the way you indicate statistical significance between groups. For understanding, it is necessary to indicate with a dash which groups are being compared, and indicate the level of significance with asterisks, and not just generally that p is less than 0.05.
Reviewer 2 Report
Comments and Suggestions for Authors
The manuscript presents a comprehensive evaluation of adenoviral-vectored vaccines expressing a consensus H5 hemagglutinin (H5CC) in mice and swine, demonstrating robust systemic and mucosal immune responses. The study is well-designed, addressing a critical gap in influenza vaccine development for swine, a key intermediary host for zoonotic transmission. The data are compelling and support the potential of adenoviral platforms for cross-clade protection. However, several points require clarification or further discussion to strengthen the manuscript.
- Figure 5: The y-axis labels for HI/MNA titers should explicitly state "log2" to avoid confusion.
- Figure 10: Survival curves lack p-values for inter-group comparisons (e.g., Ad5/Ad6 IM vs. IN).
- Were the H5 challenge viruses (e.g., Bovine/2024) pre-adapted for murine pathogenicity? If so, detail the adaptation process.
- Specify the rationale for the prime-boost interval (21 days) and its alignment with swine vaccination practices.
- Discussion Limitations,The manuscript briefly mentions maternal antibody interference but does not explore its implications for neonatal swine vaccination. This is highly relevant for field applications.
- Consider citing recent studies on ADCC/ADP in influenza protection (e.g., Zhou et al., 2021, Front Immunol).
- A supplemental figure showing phylogenetic relationships of H5CC to contemporary strains (e.g., 2024 cattle isolates) could enhance clarity.
- The failure of FluSure XP to elicit cross-reactive immunity is noted, but its inclusion seems tangential unless positioned as a "negative control." Clarify its relevance to the study’s objectives.
